# Identification of Three Novel Conidiogenesis-Related Genes in the Nematode-Trapping Fungus *Arthrobotrys oligospora*

**DOI:** 10.3390/pathogens11070717

**Published:** 2022-06-23

**Authors:** Xiaoying Liu, Qiao Miao, Zong Zhou, Siyi Lu, Juan Li

**Affiliations:** State Key Laboratory for Conservation and Utilization of Bio-Resources in Yunnan, Yunnan University, Kunming 650091, China; liuxiaoying@mail.ynu.edu.cn (X.L.); miaoqiao@mail.ynu.edu.cn (Q.M.); zhouz@mail.ynu.edu.cn (Z.Z.); lusiyi0713@mail.ynu.edu.cn (S.L.)

**Keywords:** fungus–nematode interaction, conidium yield, trap formation, stress response

## Abstract

For filamentous fungi, conidiogenesis is the most common reproductive strategy for environmental dispersal, invasion, and proliferation. Understanding the molecular mechanisms controlling conidiation and increasing conidium yield may provide promising applications in commercial development in the future for nematode-trapping fungi. However, the molecular mechanism for regulating conidium production of filamentous fungi is not fully understood. In this study, we characterized three novel conidiogenesis-related genes via gene knockout in *A. oligospora*. The absence of the genes *AoCorA* and *AoRgsD* caused significant increases in conidia production, while the absence of *AoXlnR* resulted in a decrease in conidiogenesis. Moreover, we characterized the ortholog of *AbaA*, a well-known conidiogenesis-related gene in *Aspergillus nidulans*. The deletion of *AoAbaA* not only completely abolished conidium production but also affected the production of nematode-trapping traps.

## 1. Introduction

For decades, plant-parasitic nematodes have caused terrible damage to economic crops across the world [1]. With the increasing ban of chemical nematicides, there is an urgent need to develop biocontrol agents with high efficiencies to control plant-parasitic nematodes. In soil ecology, there is a special type of fungi called nematode-trapping fungi (NTFs) which can produce nematode-trapping devices (traps) such as adhesive networks, adhesive branches, adhesive knobs, constricting rings, etc. to capture nematodes [2]. Despite showing promising applications in agriculture, the development of NTFs is still restricted as the molecular mechanisms of the NTFs interacting with nematodes are poorly understood.

Among NTFs, *Arthrobotrys oligospora*, a model nematode-trapping fungus that produces adhesive networks to capture nematodes, is the most studied fungus at present. The genome of *A. oligospora* was sequenced in 2011 [3]. As trap production is closely correlated with nematode predation ability, studies have focused on how this fungus shifts its lifestyle from saprophytic to pathogenic to capture nematodes [4,5]. This fungus shows a promising agricultural utilization potential as it would be an alternative to chemical nematicides to control plant-parasitic nematodes in the soil. A better understanding of the molecular interaction upon fungus–nematode interaction would help in the development of efficient biocontrol agents. Moreover, with *A. oligospora*, the yield of conidia is always low and the conidia are sensitive to soil environmental factors, which can reduce its predictability in field applications for controlling nematodes. Therefore, a better understanding of the molecular mechanisms of conidiation with *A. oligospora* would help to increase the conidium yield through genetic manipulation, benefitting the development of biocontrol agents to control parasitic nematodes.

For filamentous fungi, conidiogenesis is an important strategy in resisting diverse environmental stresses and in dispersing to new habitats. At present, the genetic regulation of conidiogenesis has been extensively studied in the model fungi *Neurospora crassa* and *Aspergillus nidulans*, providing an in-depth understanding of the biological processes in the asexual development of fungi [6,7]. Three key regulators, BrlA, AbaA, and WetA, are considered to comprise the central regulatory pathway in asexual development [7]. The C2H2 zinc finger transcription factor BrlA governs early development by regulating genes including AbaA, RodA, and yA [8]. WetA, the expression of which is induced by AbaA in the late stage of conidiation, then activates the expression of proteins involved in the synthesis of conidium wall components which are required for conidial maturation [9,10]. Many regulatory genes including negative regulators, upstream activators, and light-responsive genes have also been identified to be involved in conidiogenesis [6,7]. Inside the central regulatory pathway in *A. oligospora*, only the homolog of *AbaA* was found in its genome, while the homologs of *brlA* and *wetA* were not found. This suggests that the regulatory mechanisms of conidiation in *A. oligospora* may be different from those of *N. crassa* and *A. nidulans*. 

Thus, finding novel genes related to conidiogenesis in *A. oligospora* would provide valuable information for us to understand the regulatory mechanisms of conidiogenesis in *A. oligospora*. A previous study revealed that the G-protein signaling (RGS) protein RgsD is involved in the asexual development of *Aspergillus fumigatus* [11]; the homolog of RgsD in *A. oligospora* (AoRgsD) could show a similar function. In addition, whether the transportation of metal elements and the metabolism of cellulose and hemicellulose are involved in the conidiogenesis in *A. oligospora* remains an open question. Thus, AoCorA, the homolog of the membrane protein CorA whose primary function is to transport magnesium, nickel, and cobalt to maintain the homeostasis of this divalent cation in cells [12], and AoXlnR, the homolog of XlnR that mediates cellulose and hemicellulose metabolism in many other filatmentous fungi [13,14,15] were chosen as the objects of study here. The absence of *AoCorA* and *AoRgsD* caused significant increases of conidia production, while the absence of *AoXlnR* resulted in a decrease in conidiogenesis. In addition, the disruption of *AoAbaA* abolished the capacity for conidiation. This study provides novel insights to understand the mechanisms of conidiation in *A. oligospora*.

## 2. Results

### 2.1. Identification of AoAbaA, AoXlnR, AoCorA, and AoRgsD in A. oligospora

The ORF (open reading frame) of *AoAbaA* of *A. oligospora* consists of 1767 bp nucleotides with two introns and is predicted to encode a protein of 588 amino acids. As shown in Figure 1A, AoAbaA contains a TEA/ATTS domain (38 to 103 aa). 

The ORF of *AoCorA* of *A. oligospora* consists of 1878 bp nucleotides with four introns and is predicted to encode a protein of 625 amino acids. As shown in Figure 1A, AoCorA contains a MIT_CorA-like (153 to 501 aa) domain belonging to the *Thermotoga maritima* CorA (TmCorA)-family of the MIT superfamily. 

The ORF of *AoXlnR* of *A. oligospora* consists of 2979 bp nucleotides with one intron and is predicted to encode a protein of 992 amino acids. As shown in Figure 1A, AoXlnR contains a GAL4-like zinc binuclear cluster (79 to 134 aa) and a fungal specific transcription factor domain called fungal_TF_MHR (380 to 652 aa).

The ORF of *AoRgsD* of *A. oligospora* consists of 1065 bp nucleotides with no introns and is predicted to encode a protein of 354 amino acids. As shown in Figure 1A, AoRgsD contains an RGS domain (78 to 199 aa). 

### 2.2. Phenotypes of AoAbaA, AoXlnR, AoCorA, and AoRgsD Deletion Mutants

In this study, the encoding genes *AoAbaA*, *AoXlnR*, *AoCorA*, and *AoRgsD* in *A. oligospora* were respectively replaced with hph via a homologous recombination method (Figure 1B). Briefly, the 5′- and 3′-flanking fragments of each encoding gene were amplified from the genome of *A. oligospora* using corresponding primer pairs (Appendix A) and the hygromycin-resistance gene cassette (hph) was amplified from the plasmid pCSN44. The plasmid pRS426 was linearized with EcoRI and XhoI. Then, the linearized pRS426 and the three purified fragments were co-transformed into Saccharomyces cerevisiae using electroporation. The recombinant plasmids were individually transformed into *A. oligospora* protoplasts. The transformants were identified using the primers to confirm whether the encoding genes had been disrupted correctly. Southern blot analysis was used to further investigate the differences between the wild-type strain and the transformants (Figure 1C).

### 2.3. Effects of ΔAoAbaA, ΔAoXlnR, ΔAoCorA, and ΔAoRgsD Deletion Mutations on Hyphal Growth and Responses to Different Stresses

The hyphal growth rates of the WT and Δ*AoAbaA*, Δ*AoXlnR*, Δ*AoCorA*, and Δ*AoRgsD* mutant strains were compared on potato dextrose agar (PDA), tryptone-glucose (TG), and tryptone yeast-extract glucose agar (TYGA) media, respectively. However, the radial growth and the morphology showed no apparent differences between the wild-type strain and the mutants on different media (Figure 2A,B). In addition, the growth rate of the WT and mutants were compared on TG medium with different stress factors. No obvious differences were observed between the growth of the mutants and the wild-type strain on media with SDS, H_2_O_2_, and NaCl at different concentrations (Appendix A). This indicated that the wild-type and mutant strains were similar in their resistance to different stresses.

### 2.4. AoAbaA Is Also Involved in Trap Formation in A. oligospora

For nematode-trapping fungi, the capacity to produce traps is crucial for them to capture nematodes. Thus, we also compared the trap numbers among the mutant strains by adding nematodes. The WT and mutant strains were incubated on WA medium at 28 °C for 3 days. Then, approximately 200 nematodes were added to induce the formation of adhesive networks. Our results showed that all the mutant strains produced traps with the addition of nematodes. However, the number of traps in the Δ*AoAbaA* mutant was significantly reduced after adding the nematodes for 48 h, while no obvious difference was observed from the other three mutants over the same period (Appendix A).

Additionally, we counted the number of nematodes that were captured and killed by these strains. Our results showed that after 36 h, 63.9%, 57.0%, 48.7%, 39.3%, and 76.8% of the nematodes had survived in the WT, Δ*AoXlnR*, Δ*AoCorA*, Δ*AoRgsD*, and Δ*AoAbaA* strains, respectively (Appendix A).

### 2.5. AoAbaA, AoXlnR, AoCorA, and AoRgsD Were Required for Conidiation

The WT and each mutant strain were incubated on CMY medium at 28 °C for 14 days to allow conidiation. Our analyses identified that the deletion of *AoCorA* and *AoRgsD* resulted in a significantly enhanced number of conidiophores. In the WT strain from the 14-day-old cultures, the averaged conidial yields were quantified at 4.6 × 10^5^ conidia cm^−2^, whereas the conidial yields of Δ*AoCorA* and Δ*AoRgsD* were quantified at 8.1 × 10^5^ and 8.59 × 10^5^ conidia cm^−2^, respectively (Figure 3A,B). However, conidiation was completely abolished with the disruption of the *AoAbaA* gene, and significantly reduced in the absence of the *AoXlnR* gene. The average conidial yield from Δ*AoXlnR* was 2.15 × 10^5^, indicating a 53% decrease in conidiation capacity in the absence of *AoXlnR* (Figure 3A,B).

Although the conidiation capacities of the WT and the mutants were different, their conidial morphology showed no obvious differences (Figure 3C). Furthermore, the conidial germination rates of the mutants showed no significant difference with the WT strain on WA media (Figure 3D,E).

### 2.6. Relationship between the Expressions of AoAbaA, AoXlnR, AoCorA, and AoRgsD and Conidiation

To determine whether *AoXlnR*, *AoCorA*, and *AoRgsD* play a crucial role during the conidiation of *A. oligospora*, we investigated the expression levels of these four genes at 2, 4, 6, and 8 days. Our results showed that the expression of all four genes showed a significant increase at 6 days (Figure 4A). The expression of *AoAbaA* was down-regulated at 8 days (Figure 4D), but *AoXlnR*, *AoCorA*, and *AoRgsD* retained a high expression at 8 days (Figure 4A–C). 

To understand the regulatory relationships among these four genes, we also compared the transcript levels of these four genes in the other three mutants. Results showed that the transcript level of *AoAbaA* was enhanced in Δ*AoXlnR*, Δ*AoCorA*, and Δ*AoRgsD* at 2 and 8 days, but showed no obvious changes in Δ*AoXlnR* and Δ*AoCorA* at 4 and 6 days though the Δ*AoRgsD* strain’s expression was still enhanced (Figure 4E). For *AoXlnR*, no obvious transcript change was observed, suggesting that *AoRgsD* and *AoCorA* don’t regulate the expression of *AoXlnR* (Figure 4F). For *AoCorA*, the transcript level was enhanced in Δ*AoXlnR* but showed no obvious change in Δ*AoRgsD*, suggesting that *AoCorA* may be directly or indirectly regulated by *AoXlnR* (Figure 4G). For *AoRgsD*, its expression level showed no change in Δ*AoXlnR* and Δ*AoCorA* strains (Figure 4H). In conclusion, we speculated that AoAbaA was negatively regulated by *AoRgsD* and it may also be regulated by *AoXlnR* and *AoCorA* at the initial and late stages of conidiation. Moreover, *AoCorA* may be directly or indirectly regulated by *AoXlnR*.

Furthermore, the expression levels of these genes related to conidiation in *A. oligospora* were compared between the WT and mutant strains (Appendix A). In Δ*AoRgsD*, the expression of *vosA* was up-regulated at all the tested times. However, the transcript levels of *lreB*, *fluG* and *veA* were up-regulated at 2 days, but showed no obvious change at the other three tested times. The expression of *pkA* was up-regulated at the last stage of conidiation (6 and 8 days). In Δ*AoXlnR*, *flbA*, *vosA*, *lreA*, *lreB*, *pkA*, *pkaR*, and *pkaC1* were up-regulated at all the tested times, while *veA*, *fluG*, *plp1*, and *velB* were up-regulated during the initial days but showed no changes at the later stage. In Δ*AoCorA*, the expressions of *vosA*, *pkaR*, and *pkaC1* were up-regulated, and *fluG* was up-regulated at 2 days but exhibited no change at the other three tested time points. At 8 days, the expressions of *veA*, *velB*, *lreA*, and *nsdD* were up-regulated. In addition, *flbA*, *plp1*, *nsdD*, *pKA*, and *p38 MAPK* were down-regulated in Δ*AoCorA* at 2 days.

## 3. Discussion

*A**. oligospora* is a prominent predatory fungus that can capture nematodes with its specific trap devices. Recently, scientists have mainly focused on studying the molecular mechanisms of trap formation. However, for a filamentous fungus, asexual development is also important for this fungus to survive in various environments. Although previous studies have confirmed that several versatile genes such as the autophagy-related genes (*Aoatg1*, *Aoatg4*, *Aoatg5*, and *Aoatg13*) [16,17,18], transcription factor-encoding genes (*AoStuA* and *APSES*) [19], genes involved in the protein kinase signaling pathway (*Slt2*, *Fus3*, and *Bck1*, and *Ime2*) [20,21], and GTPases [22,23] may influence the capacity for conidiogenesis, these genes can also regulate multiple biological processes including the growth of trap formation in *A. oligospora*. In this study, we functionally characterized three genes (*AoCorA*, *AoRgsD*, and *AoXlnR*) that may specifically regulate conidiogenesis in *A. oligospora*, as the disruption of these three genes neither influences the growth nor the capacity for resistance to different stress and trap formations. However, Δ*AoCorA* and Δ*AoRgsD* deletion mutants showed significant increases in conidia production, while the absence of *AoXlnR* resulted in a decrease in conidiogenesis (Figure 3). Thus, we believe that these three genes may perform specific functions during conidiation in *A. oligospora*. 

The RGS domain is an essential part of the ‘regulator of G-protein signaling (RGS)’ protein family, members of which play critical regulatory roles as GTPase-activating proteins (GAPs) of the heterotrimeric G-protein G-alpha-subunits [24]. In Gram-positive bacteria, the RgsD protein shows a similar structure to the ancillary pilin RrgC from *Streptococcus pneumoniae*, which is a surface-associated protein that likely participates in covalent adhesion [7]. A previous study implied that RgsD might play a role in the positioning of PG synthesis factors or the regulation of their activity [7]. In filamentous fungi, RGSs regulate diverse signals that control vegetative growth, sporulation, stress responses, secondary metabolism, and virulence [25]. In the human pathogenic fungus *Aspergillus fumigatus*, RgsD is one of the six regulators of G-protein signaling (RGS) proteins (FlbA, GprK, RgsA, Rax1, RgsC, and RgsD), which have been identified to negatively control development, toxigenesis, stress response, and virulence [11]. A previous study on *A. fumigatus* revealed that RgsD negatively down-regulates cAMP-PKA signaling pathway to control asexual development, as conidia numbers were drastically increased in the absence of *RgsD* [11]. In this study, the expression of PKA was up-regulated at the late stage of conidiation (6 and 8 days) in *A. oligospora* and Δ*AoRgsD* produced a higher number of conidiophores than WT, which is consistent with the study in *A. fumigatus* (Figure 3 and Figure 4E). Thus, we believe that RgsD has a conserved function in conidiation in filamentous fungi. Moreover, the mRNA levels of *AoAbaA* and *vosA* were significantly increased at all time points tested in Δ*AoRgsD* (Figure 4E and Appendix A), suggesting that *AoAbaA* and *vosA* may also be negatively regulated by *AoRgsD* during conidiation.

XlnR, a binuclear zinc-finger transcription factor belonging to the GAL4 superfamily, is responsible for the transcription of xylanolytic genes (*xlnB*, *xlnC*, and *xlnD*) [26], endoglucanase (*eglA* and *eglB*), and cellobiohydrolase genes, and some galactosidase-encoding genes. This suggests a general role for *XlnR* in cellulose and hemicellulose degradation and metabolism [13,14,15]. Previous findings have shown that the deletion of the C-terminus of *XlnR* results in an increase in xylanase activity [27]. In *F. oxysporum*, *XlnR* was not an essential virulence determinant [28]. In this study, we found that AoXlnR contains a GAL4-like zinc binuclear cluster and a fungal_TF_MHR domain, which has been suggested to assist the C6 zinc cluster in DNA target discrimination [29]. The conidial yield of *A. oligospora* was significantly affected by the deletion of *AoXlnR* (Figure 3). However, RT-PCR analysis showed that the expression of *AoXlnR* was up-regulated during conidiogenesis (Figure 4C), and at least six genes (*flbA*, *vosA*, *lreA*, *lreB*, *pkA*, *pkaR*) involved in conidiogenesis were up-regulated in Δ*AoXlnR* (Appendix A), suggesting that *AoXlnR* may negatively regulate these genes related to conidiation. This is the first confirmation that *XlnR* conducts the function of conidial regulation. 

CorA is a membrane protein whose primary function is to transport magnesium, nickel, and cobalt to maintain the homeostasis of this divalent cation in cells [12]. Previous studies have shown that the MIT_CorA-like domain is an essential membrane protein involved in transporting divalent cations (uptake or efflux) across membranes [12]. In *Pectobacterium carotovorum*, a phytopathogenic bacterium that causes soft rot disease in plants, *CorA* affects virulence and extracellular enzyme production [30]. In addition, mutation of the *corA* gene in *Salmonella enterica* serovar *Typhimurium* leads to an attenuation of virulence and other defects [31]. In *Magnaporthe oryzae*, the CorA transporter is required for mycelial growth and surface hydrophobicity [32]. Intriguingly, the deletion of *AoCorA* in *A. oligospora* results in an increase in conidial yield (Figure 3), suggesting that magnesium/nickel/cobalt homeostasis is essential for conidiogenesis. RT-PCR analysis showed that the disruption of *AoXlnR* can lead to an up-regulation of *AoCorA* (Figure 4G), suggesting the negative regulation of *AoCorA* by *AoXlnR*—though their regulatory mechanism is still unclear. In addition, the transcript levels of several genes related to conidiation were also influenced in Δ*AoCorA* (Appendix A). Our research confirmed that the *AoCorA* might also be involved in the regulation of conidiation. 

In several model fungi, the three central regulators BrlA, AbaA, and WetA have been identified as playing crucial roles in asexual development [7]. Interestingly, only *AoAbaA*, the ortholog of *AbaA*, was identified in *A. oligospora*, suggesting that the genetic program of conidiation in *A. oligospora* may be different from other filamentous fungi. In our study, the disruption of *AoAbaA* resulted in the abolishment of conidiogenesis and a reduction in trap formation (Figure 3). Furthermore, the survival rate of *Caenorhabditis elegans* in Δ*AoAbaA* was higher than that of the wild-type strain, suggesting that there may be a common pathway where *AoAbaA* is involved in both conidiation and trap formation. Although the yield of traps showed no obvious differences among the WT and Δ*AoXlnR*, Δ*AoCorA*, Δ*AoRgsD*, and Δ*AoAbaA* strains, the survival rates of *C. elegans* in Δ*AoCorA* and Δ*AoRgsD* were significantly lower than in the WT strain (Appendix A), implying that the conidial yields might influence the ability to capture nematodes.

In conclusion, the regulation mechanism for sporulation has both similar and unique features among fungal species. Here, we characterized three novel genes that were involved in conidiation in the model nematode-trapping fungus *A. oligospora*. *AoRgsD* regulates conidiation through the PKA/VosA/abaA pathway (Figure 5). *AoXlnR* may directly or indirectly affect conidiation via *AoCorA* and/or the metabolisms of cellulose, xylanase, and hemicellulose. *AoXlnR* may also negatively regulate the PKA signaling pathway, as the expression of *AopkaA* was enhanced in Δ*AoXlnR* (Figure 5). In addition, the magnesium, nickel, and cobalt transported by *AoCorA* may also play a key role during conidiation (Figure 5). Together, our study provides novel insights into the regulation of conidiation in nematode-trapping fungus *A. oligospora*.

## 4. Materials and Methods

### 4.1. Fungal Strains and Culture Conditions

The wild-type strain of *A. oligospora* (ATCC24927) was cultured on potato dextrose agar (PDA) at 28 °C. *S. cerevisiae* (FY834) was grown in YPD (yeast extract peptone dextrose) plates. The *Escherichia coli* strain DH5a (Takara, Shiga, Japan) was used to keep the plasmid pRS426 [33]. The medium PDASS (PDA supplemented with 10 g/L molasses and 0.4 M saccharose) with the addition of 200 µg/mL hygromycin B (Amresco, Solon, OH, USA) was used for selecting transformants. *C. elegans* nematodes were used in this study, which were grown on oatmeal agar medium at 26 °C and separated using the Baerman funnel technique [34].

### 4.2. Identification of AoAbaA, AoXlnR, AoCorA, and AoRgsD in A. oligospora

The encoding genes of *AoAbaA*, *AoCorA*, *AoXlnR*, and *AoRgsD* (GenBank no. *XM_011124438*, *XM_011126761*, *XM_011119393*, and *XM_011129373*, respectively) were identified from the *A. oligospora* genome. The conserved functional domain structures of these four proteins were predicted using the InterProScan 5.0 (http://www.ebi.ac.uk/interpro/, accessed on 8 April 2022) with default parameter settings.

### 4.3. Deletion of the Genes AoAbaA, AoXlnR, AoCorA, and AoRgsD

Genomic DNA of the fungus *A. oligospora* was extracted using the CTAB (cetyl trimethylammonium bromide) procedure. The encoding genes of *AoAbaA*, *AoCorA*, *AoXlnR*, and *AoRgsD* were knocked out following the method previously described (Figure 1B) [35]. Briefly, the 5′- and 3′-flanking fragments of each encoding gene were amplified from the genome of *A. oligospora* using corresponding primer pairs (Appendix A). The hygromycin-resistance gene cassette (*hph*) was amplified from the plasmid pCSN44. The plasmid pRS426 was linearized with E*coR*I and X*ho*I. The linearized pRS426 and the three purified fragments were then co-transformed into *S. cerevisiae* using electroporation. The recombinant plasmids were individually transformed into *A. oligospora* protoplasts. The transformants were identified using the primers to confirm whether the encoding genes had been disrupted correctly. Furthermore, Southern analysis was carried out in accordance with our previous reports (Figure 1C) [36]. The primers used to prepare Southern hybridization probes and the restriction enzymes used for different genes are listed in Appendix A.

### 4.4. Comparison in Growth Rate between Mutants and Wild-Type Strain in Different Conditions

The WT and Δ*AoAbaA*, Δ*AoCorA*, Δ*AoXlnR*, and Δ*AoRgsD* mutants with similar-sized colonies were incubated on PDA, TYGA (10 g/L tryptone, 5 g/L yeast extract, 10 g/L glucose, 5 g/L molasses, 20 g/L agar), CMA (20 g/L maizena, 20 g/L agar), and TGG (10 g/L tryptone,10 g/L glucose, 20 g/L agar) medium at 28 °C for 5 days, respectively. The growth rate and colony morphology were observed and measured daily until mycelia covered the whole surface of the 6 cm diameter disks.

To compare their stress response capacity, mycelial plugs of each strain were inoculated onto TG medium with NaCl (0.1–0.3 M), H_2_O_2_ (5–15 mM), and SDS (sodium dodecyl sulfate) (0.02–0.03%) at 28 °C for 5 days, respectively. The diameters of the colonies were measured every day.

### 4.5. Comparison of Conidial Production, Morphology, and Germination

The fungal plugs (7 mm in diameter) of the WT and mutant strains were cultivated on CMY culture medium at 28 °C for 10 days. The conidia on each plate were washed with 5 mL of sterile water, then mycelial debris was removed using four layers of lens tissues to filtrate. The number of conidia per square millimeter of each colony was determined by hemocytometer [37]. Meanwhile, conidiophores were observed under a light microscope (BX51, Olympus, Tokyo, Japan). To calculate the conidial germination rates, 50-μL aliquots of 10^5^ conidia mL^−1^ suspensions of the WT strain and mutants were used to inoculate CMY medium (20 g/L maizena, 5 g/L yeast extract, 20 g/L agar). Samples were incubated at 28 °C, and the germinated conidia were counted at 4-h intervals until 24 h post inoculation.

### 4.6. Trap Formation and Pathogenicity Assays

To characterize the capacity for trap formation in the *A. oligospora* WT strain and mutants, approximately 10^4^ conidia of the wild-type strain and the Δ*AoCorA*, Δ*AoXlnR* and Δ*AoRgsD* mutants were spread on WA plates and maintained at 28 °C for 5 days. Meanwhile, Δ*AoAbaA* was incubated on WA medium using colony plugs of 5 mm diameter, as this strain cannot produce conidia. Then, 200–300 nematodes were added to each plate to induce trap formation. The traps were counted under a light microscope (BX51, Olympus, Tokyo, Japan) at specified time points.

### 4.7. Quantitative Real-Time PCR (RT-PCR) Analysis

To determine the transcript levels of genes involved in sporulation, the WT strain and mutants were incubated on CMY medium at 28 °C for 2, 4, 6, and 8 days, and hyphae were collected for RT-PCR analysis. The total RNA of all samples was extracted using the RNeasy kit (QIAGEN Inc., Hilden, Germany) according to the manufacturer’s instructions. First-strand cDNA was synthesized using the Superscript III First-Strand cDNA Synthesis SuperMix kit (Invitrogen, Carlsbad, CA, USA). The constitutively expressed housekeeping gene β-tubulin (Tub, *AOL_s00076g640*) was used as the internal standard for the normalization of expression of the other genes studied. Three cDNA samples were analyzed under the action of a SYBR^®^ Premix Ex Taq™ (TaKaRa). Each sample was run in triplicate and two biological replicates were employed. The 2^−ΔΔCt^ method was used to measure the relative transcript levels of each gene [38]. The primers used were designed using Primer 5 and are listed in Appendix A.

### 4.8. Statistical Analysis

Each experiment was performed with three biological replicates and the data from individual treatments are expressed as mean ± SD. GraphPad Prism software, version 6 (GraphPad Software Inc., San Diego, CA, USA) was used for the photographs and statistical analyses (one-way ANOVA). *p* < 0.05 was considered to indicate significant differences.

## Figures and Tables

**Figure 1 pathogens-11-00717-f001:**
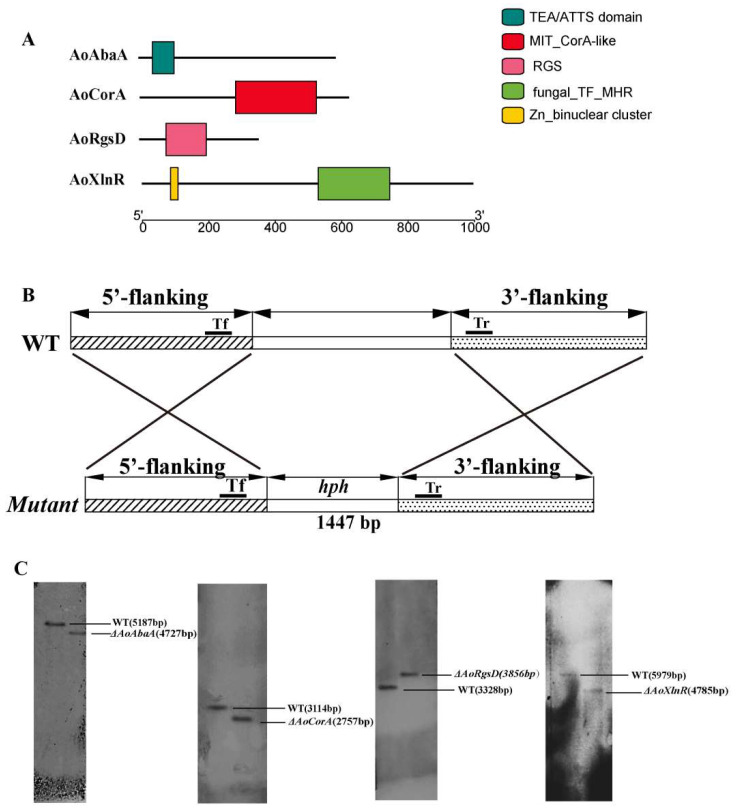
Protein domain structure and deletion of targeting genes in *A. oligospora.* (**A**) Protein domain structures of AoAbaA, AoXlnR, AoCorA, and AoRgsD, respectively; (**B**) the sketch map of replacement of targeting genes using the homologous recombination method; (**C**) Southern blot analysis of the wild-type (WT) and mutant strains.

**Figure 2 pathogens-11-00717-f002:**
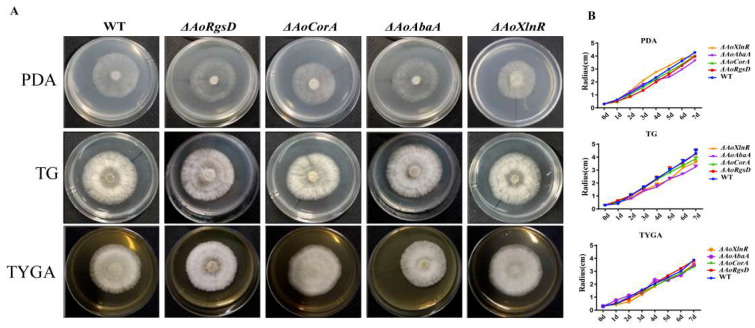
Comparison of the growth rate and colony morphology of the wild-type strain and mutants. (**A**) The colony morphology of *A. oligospora* wild-type strain and mutants on PDA, TG, and TYGA media for 3 days at 28 °C; (**B**) comparison of the growth rate of the WT and mutants on PDA, TG, and TYGA media.

**Figure 3 pathogens-11-00717-f003:**
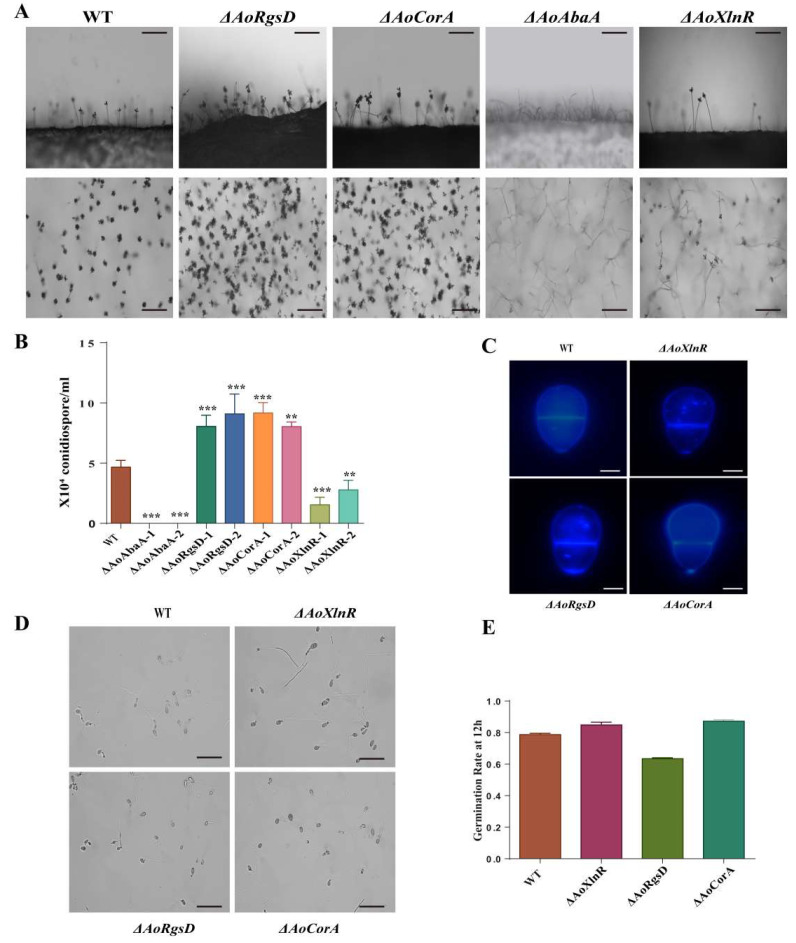
Comparison of the sporulation, spore germination, and conidia morphology of the wild-type strain and mutants. (**A**) Conidiophore differentiation in the WT and mutant lines. Bar = 100 um; (**B**) sporulation of the wild-type strain and mutants. ** *p* < 0.05, *** *p* < 0.001 versus WT (wild type); (**C**) comparison of the conidia morphology. Bar = 30 um; (**D**) comparison of the spore germination rate between the wild-type strain and mutants. Bar = 100 um; (**E**) statistical results of the germination rate of the wild-type strain and mutant.

**Figure 4 pathogens-11-00717-f004:**
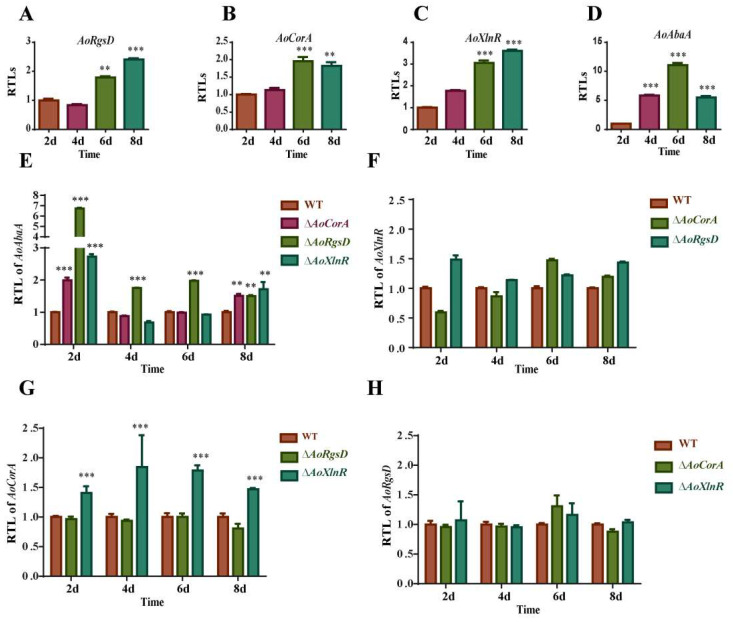
Expression of genes in the wild-type strain and mutants during sporulation. (**A**) The relative transcriptional levels (RTLs) of the gene *AoRgsD* during vegetative growth and conidiation (2, 4, 6, and 8 days) in *A. oligospora*; (**B**) the RTLs of the gene *AoCorA* during vegetative growth and conidiation (2, 4, 6, and 8 days) in *A. oligospora*; (**C**) the RTLs of the gene *AoXlnR* during vegetative growth and conidiation (2, 4, 6, and 8 days) in *A. oligospora*; (**D**) the RTLs of the gene *AoAbaA* during vegetative growth and conidiation (2, 4, 6, and 8 days) in *A. oligospora*; (**E**) the RTLs of the gene *AoAbaA* in the Δ*AoXlnR*, Δ*AoCorA*, and Δ*AoRgsD* strains, respectively; (**F**) the RTLs of the gene *AoXlnR* in the Δ*AoCorA*, and Δ*AoRgsD* strains, respectively; (**G**), the RTLs of the gene *AoCorA* in the Δ*AoXlnR* and Δ*AoRgsD* strains, respectively; (**H**) the RTLs of the gene *AoRgsD* in the Δ*AoXlnR* and Δ*AoCorA* strains, respectively. *** *p* < 0.001, ** *p* < 0.05.

**Figure 5 pathogens-11-00717-f005:**
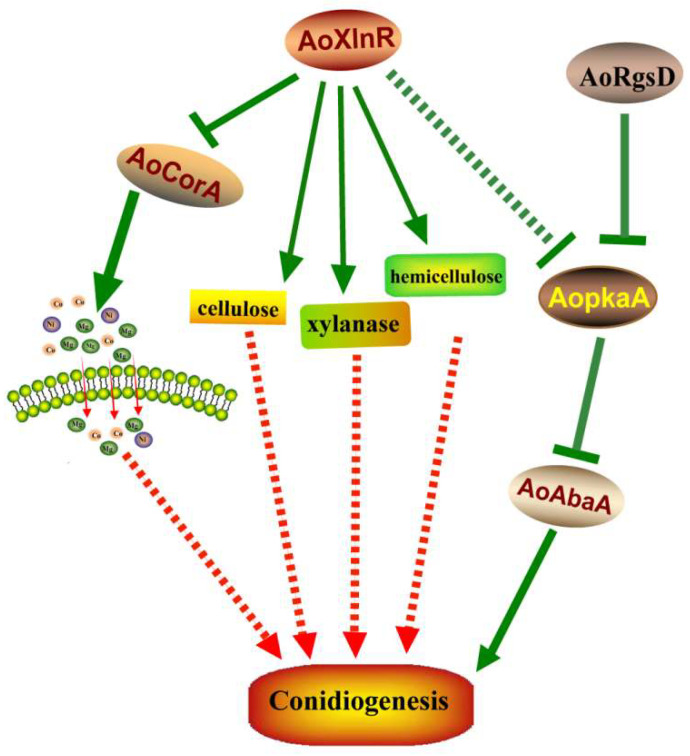
A proposed model for the regulation of *AoAbaA*, *AoXlnR*, *AoCorA*, and *AoRgsD* in *A. oligospora.* Our findings suggest that *AoRgsD* regulates conidiation through the PKA/VosA/abaA pathway. *AoXlnR* may directly or indirectly affect conidiation via *AoCorA* and/or the metabolisms of cellulose, xylanase, and hemicellulose. *AoXlnR* may negatively regulate the PKA signaling pathway.

## Data Availability

Not applicable.

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
