# Peer review of "Identification of Three Novel Conidiogenesis-Related Genes in the Nematode-Trapping Fungus Arthrobotrys oligospora"

_pathogens, 2022, doi:10.3390/pathogens11070717_

Round 1

Reviewer 1 Report

The manuscript ‘Identification of three novel conidiogenesis-related genes in the nematode-trapping fungus Arthrobotrys oligospora’ by Xiaoying Liu et al describes the identification of three A. oligospora genes which are influencing conidiation in that nematode trapping fungus. Though several experiments are conducted and described well, several issues should be addressed.

Main points:

-       Unfortunately, I was not able to find information why the authors have choosen to delete specifically those genes described in the study. Can you please explain?

-       I find the conclusion about the interaction of the gene products in conidiation rather speculative and thus invite the authors to support these with further experiments. For example, the generation of double deletion mutants would be a nice tool to show that the genes are present in the same genetic pathway. This would be especially of interest for the genes AoRgs and AOCorA, which have similar conidiation patterns.

-       To prove that the conducted gene deletions do cause the phenotypes as stated, it is crucial to generate complementation strains, which harbor again the native genes. These should behave wildtype-like.

Minor points:

-       Introduction: To explain the rational of the study better, the paragraph explaining the role of conidiation for the application of A. oligospora as biocontrol agent (line 57-62) could be integrated just after the last sentence of line 41.

-       Please revise the formatting of the manuscript and ensure that organisms and genes are written in italics throughout the text.

-       Figure 1: It would improve the readability of the manuscript when the different genes displayed in Figure 1 part A would follow the same chronological order as the corresponding information in the text.

-       Figure 1: part C: please increase the contrast of the southern blot displayed, the lines are hard to see. Further, you should include the marker lanes for all southern blots provided.

-       Figure 2: Figure parts B and C are very small and I am unable to read the information given. Further, I suggest to move B and C to the supplement since no differences are shown for all growth conditions and strain tested.

-       Figure 3: Figure parts B and E: please enlarge, I am not able to read the information given. Figure part D, legend: here, no germination rate is displayed, please revise the legend. I further miss the size bars in A and D.

-       Structure: Please rearrange the structure of the results section. At the moment, part 2.5 about trap formation splits the rest of the manuscript which talks about conidiation.

-       Figure 4: again, the graphs are too small to be read.

-       Material and Methods: in several paragraphs the amount of cells used is not displayed well. Please adapt.

Reviewer 2 Report

Dear authors,

you´ll find my comments in the attachment.

Best wishes

Reviewer 3 Report

Please find my comments in the attached pdf file.

Good luck

Round 2

Reviewer 1 Report

The manuscript ‘Identification of three novel conidiogenesis-related genes in the nematode-trapping fungus Arthrobotrys oligospora’ by Xiaoying Liu et al describes the identification of three A. oligospora genes which are influencing conidiation in that nematode trapping fungus. The authors have improved the quality of the manuscript significantly in the revised version and have provided important information about experimental limitations. The only remaining point from my side is the quality of Figure 1 Part B. Still, the graphs are too small to be read properly. Please increase size. 
